# *Brettanomyces bruxellensis* wine isolates show high geographical dispersal and long persistence in cellars

Alice Cibrario[1], Marta Avramova[1], Maria Dimopoulou[1,2], Maura Magani[1], Cécile Miot-Sertier[1], Albert Mas[3], Maria C. Portillo[3], Patricia Ballestra[1], Warren Albertin[1,4]*, Isabelle Masneuf-Pomarede[1,5], Marguerite Dols-Lafargue[1,4]

**1** Univ. Bordeaux, ISVV, Unité de recherche Œnologie EA 4577, USC 1366 INRA, Bordeaux INP, Villenave d'Ornon, France, **2** Department of Food Science and Technology, Faculty of Agriculture, Forestry and Natural Environments, Aristotle University of Thessaloniki, Thessaloniki, Greece, **3** Biotecnología Enológica. Dept. Bioquímica i Biotecnologia, Facultat d'Enologia. Universitat Rovira i Virgili. C/ Marcel·lí Domingo, Tarragona, Spain, **4** ENSCBP, Bordeaux INP, Pessac, France, **5** Bordeaux Sciences Agro, Gradignan, France

* warren.albertin@u-bordeaux.fr

## Abstract

*Brettanomyces bruxellensis* is the main wine spoiler yeast all over the world, yet the structure of the populations associated with winemaking remains elusive. In this work, we considered 1411 wine isolates from 21 countries that were genotyped using twelve microsatellite markers. We confirmed that *B. bruxellensis* isolates from wine environments show high genetic diversity, with 58 and 42% of putative triploid and diploid individuals respectively distributed in 5 main genetic groups. The distribution in the genetic groups varied greatly depending on the country and/or the wine-producing region. However, the two possible triploid wine groups showing sulfite resistance/tolerance were identified in almost all regions/countries. Genetically identical isolates were also identified. The analysis of these clone groups revealed that a given genotype could be isolated repeatedly in the same winery over decades, demonstrating unsuspected persistence ability. Besides cellar residency, a great geographic dispersal was also evidenced, with some genotypes isolated in wines from different continents. Finally, the study of old isolates and/or isolates from old vintages revealed that only the diploid groups were identified prior 1990 vintages. The putative triploid groups were identified in subsequent vintages, and their proportion has increased steadily these last decades, suggesting adaptation to winemaking practices such as sulfite use. A possible evolutionary scenario explaining these results is discussed.

## Introduction

*Brettanomyces bruxellensis* is one of the most infamous wine spoiler yeast, able to contaminate up to 25% of red wines [1, 2]. Indeed, *B. bruxellensis* is known to produce specific compounds like volatile phenols, associated with unpleasant aromas, usually described as "horse sweat" or

**Funding:** This work received financial support from the Conseil Interprofessionel des Vins de Bordeaux (CIVB, Grant number: 2014/2015 40792), from Région Aquitaine (Grant number: 2014:1R20203-00002990), from France Agrimer (Grant number: 7120154146) and by the French National Research Agency (ANR-18-CE20-0003). A.M. and M.C.P. participation was supported by a project from the Spanish Government (AGL2015-73273-JIN). The funders had no role in study design, data collection and analysis, decision to publish, or preparation of the manuscript.

**Competing interests:** The authors have declared that no competing interests exist.

"leather" [3]. The contaminated wines have tainted organoleptic perception, decreased fruitiness [4, 5] and consequently are rejected by the consumers [6].

An important bibliography is dedicated to the *B. bruxellensis* species, with 100 to 200 papers published each year over the last decade (source: Google Scholar). Many papers investigate volatile phenol production [7–10], the biotic and abiotic factors impacting *B. bruxellensis* growth [11–15] and some peculiarities of the species like the ability to survive in the VNC (Viable Non Culturable) state [7, 16, 17] or the specific oxygen needs during fermentation [18]. Moreover, different detection and quantification methods for *Brettanomyces*, ranging from direct plating methods through molecular detection and flow cytometry analysis, were examined (see Tubia et al., 2018 for review [19]). The genetic diversity of the species has also been largely investigated, and a plethora of approaches were developed across the years, including RAPD (Random Amplified Polymorphic DNA) [20], AFLP (Amplified Fragment Length Polymorphism) [21], REA-PFGE (pulsed field electrophoresis) [22], Sau-PCR [23], PCR-DGGE [24], mtDNA restriction analysis, ISS-PCR (introns 5′ splice site sequence)[25, 26], etc. In the wine industry, most of these genetic analyses revealed high diversity within the species, at the vineyard, in the winery or at sample levels [8, 23, 27–29]. However, in most cases, only a small subset of isolates (a few dozens) were included, and these markers, although discriminant, were not appropriate for population genetic studies. Recently, a great advance was made with the genome sequencing of different *B. bruxellensis* strains [30–36], revealing the existence of diploid and allotriploid strains. This genetic oddity prompted the development of microsatellite markers that are codominant and thus can be used to assess the possible ploidy level of an individual (maximum 2 alleles per locus means possible diploid, maximum 3 alleles per locus possible triploid, etc.) [37–39]. Microsatellites are also particularly well-adapted for large-scale population studies [40, 41]. Twelve markers were applied to a unique collection of more than 1500 strains of *B. bruxellensis* from various countries and different fermentation niches (wine, beer, bioethanol, tequila, kombucha, cider) [41, 42]. The strains were clustered in 6 genetic groups, depending on both their putative ploidy level (diploid versus triploid) and their substrate of isolation [41]. Besides their genetic difference, these populations presented contrasted phenotypes: two different groups of triploid strains, mostly associated with wine substrate, showed tolerance or resistance to sulfur dioxide, the most common preservative used in winemaking [42–44]. A preliminary study on a small subset of 8 strains suggested variability in bioadhesion and colonization properties [45]. Altogether, these results indicate that the genetic diversity of *B. bruxellensis* is shaped by anthropic activities, including the winemaking process. Though, the precise impact of wine-related activities on *B. bruxellensis* populations remains to be precisely described. In this work, we focused on the 1411 isolates previously genotyped associated with wine niche (wine, grapes, cellar equipment, etc.). We searched for the geographical and temporal trends underlying wine *B. bruxellensis* diversity. Finally, a specific attention on 'clones' (i.e. isolates displaying identical genotypes) is proposed.

## Material & methods

### Yeast strains

We used 1411 isolates of *B. bruxellensis* associated with the wine production from 21 countries (S1 Table). Agar-YPD medium containing 10 g.L−1 yeast extract (Difco Laboratories, Detroit M1), 10 g.L−1 bactopeptone (Difco Laboratories, Detroit M1), 20 g.L−1 D-glucose (Sigma-Aldrich) and 20 g.L−1 agar (Sigma-Aldrich) was used for day-to-day growth. All isolates were kept at -80˚C in glycerol:YPD (50:50) medium.

## Microsatellite genotyping

Twelve microsatellites were used for *B. bruxellensis* genotyping as previously described [41]. Briefly, DNA was extracted by lysing fresh colonies in 30 μL of 20 mM NaOH (99°C, 10 minutes). Touchdown PCR were performed in a final volume of 15 μL containing 1 μL of DNA extract, 0.05 μM of forward primer, 0.5 μM of reverse primer and labelled primer, 1x Taq-&GO (MP Biomedicals, Illkirch, France). The sequences of the primers, PCR program and dilution conditions are detailed in Avramova et al. (2018). The size of PCR fragments were determined using an ABI3730 DNA analyser (Applied Biosystems) and GeneMarker Demo software V2.2.0 (SoftGenetics). More than 17 96-well microplates were needed to analyse the whole population. Thus, control strains (AWRI1499 and/or CBS 2499) were used to check for deviation between microplates and normalized the data. All genotyping analyses were performed in the same laboratory (UR Oenology) to minimize technical variation. Most of the microsatellite datasets (1488 isolates encompassing non-wine strains) were published by Avramova et al (2017), with strains addition (from Greek wines) by Dimopoulou et al (2019) and unpublished isolates from Catalonia wines [40, 41].

## Data analysis

The microsatellite dataset was analyzed using R and various packages. Principal Component Analysis (PCA) was performed using ade4 package [46]. Not available (NA) data (around 6% missing data) were replaced by the closest neighbour data (only for PCA analysis). The connection network and minimum spanning tree was built using the chooseCN function from adegenet package [47, 48]. In order to determine whether the observed clustering was purely due to the presence of 2 or 3 alleles in putative diploids and triploids respectively, the following simulation was performed: for each strain and each locus showing 3 alleles, we randomly removed one of the three alleles and performed the PCA on this randomly 2N-constrained dataset.

Diversity indexes were calculated using the poppr package [49, 50], and 95% confidence intervals were calculated using 100 bootstrap replicates.

For the geographic distribution, maps were drawn using R *maps* package and pies using *graphics* package. Kilometric distances between clones were calculated from longitude and latitude coordinates using the sp package [51].

## Results

### *B. bruxellensis* wine isolates show high genetic diversity and distribution varying with the country and the vineyard region

1411 wine isolates from 21 countries were included in our analysis, resulting in 340 genotypes. The isolates' set is represented as a minimum spanning tree (Fig 1). In a previous study that encompassed other substrates (beer, kombucha, etc), we defined 6 subpopulations that clustered depending on substrate origin and the possible ploidy level of the strains: strains having maximum 2 alleles per locus were considered as possible diploids, while those having maximum 3 alleles per locus were considered as putative triploids [41]. The six previously defined clusters were globally well conserved with this subset of wine isolates, although the position of a few isolates, located at the periphery of the clusters, seemed poorly resolved. Our wine isolates were distributed as follow (Table 1): 521 isolates belong to the so-called diploid wine group (Wine 2N, CBS 2499-like, darkcyan), 551 to a possibly triploid wine group (1st Wine 3N, AWRI1499-like, red), 229 to a possible triploid beer group (Beer 3N, AWRI1608-like, orange), 69 to the kombucha diploid group (Kombucha 2N, L14165-like, green), 40 to a

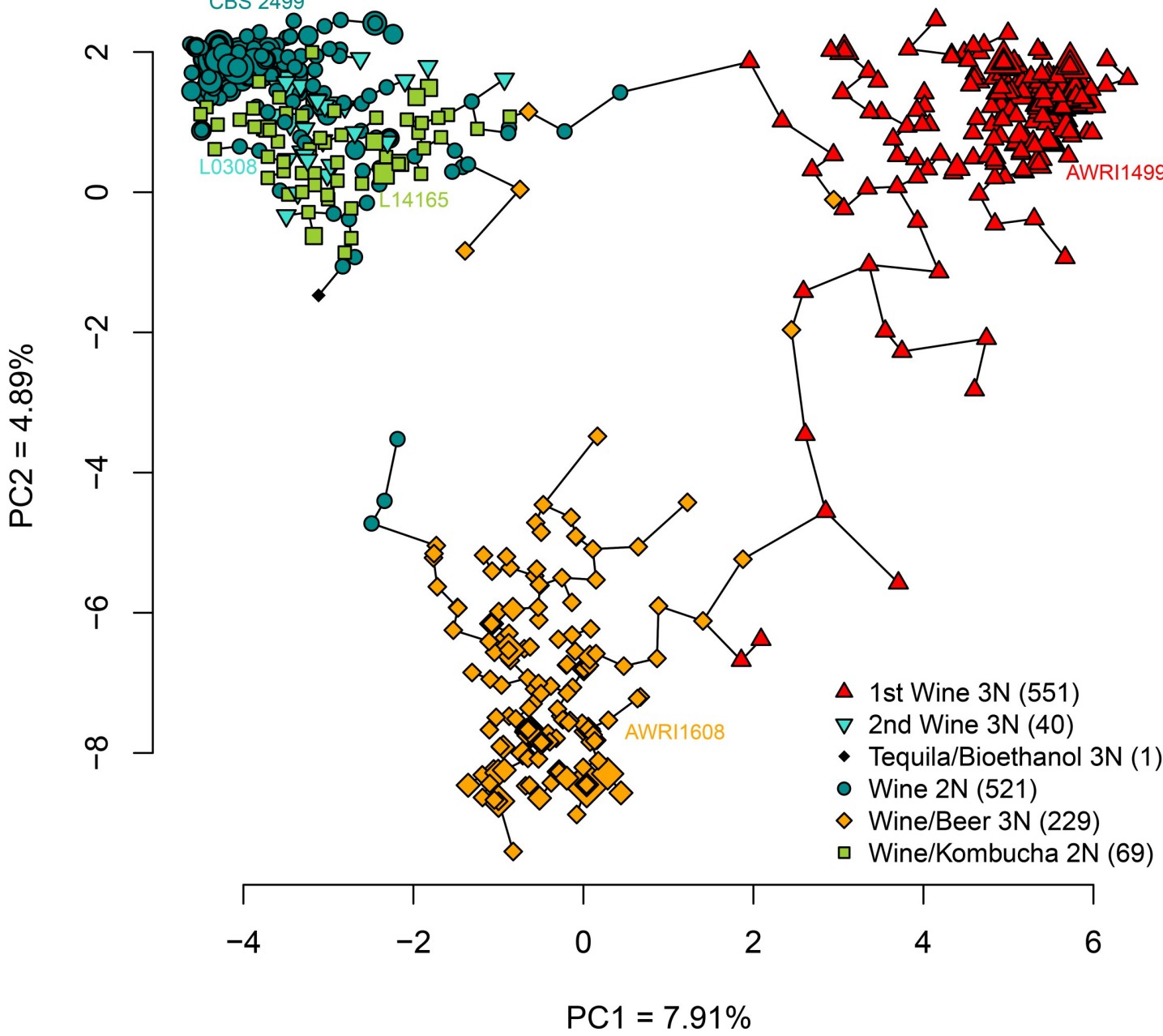

**Fig 1. Minimum spanning tree of wine *Brettanomyces bruxellensis* isolates based on genetic distances.** 1411 strains were genotyped using 12 microsatellite markers. A PCA was performed using the R ade4 package. Only the two first axes (principal component, PC1 and PC2) were represented. The connection network and minimum spanning tree was built using the chooseCN function from R adegenet package. For genetically identical isolates (aka 'clones'), the size of the points is log10 proportional to the number of isolates.

second possible triploid wine group (2nd Wine 3N, L0308-like, turquoise) and 1 from a possible triploid tequila/bioethanol group (blue). To determine whether the observed clustering was due to the presence of additional alleles for the putative triploid strains compared to the diploid ones, a similar analysis was performed on a 2N-constrained dataset (maximum 2 alleles/loci, randomly picked). The resulting minimum spanning tree (S1 Fig) was very close to the one obtained with the complete dataset, indicating that the different populations clustered depending on the quality of their alleles besides their quantity.

**Table 1. Distribution of 1411 wine isolates of *Brettanomyces bruxellensis* and main diversity parameters.**

| Group name–color | Reference strain | Number of wine isolates | Number of genotypes (richness) | Shannon's diversity index | Shannon's Equitability index (Evenness) | Simpson's diversity index | Simpson's Equitability index |
|---|---|---|---|---|---|---|---|
| Wine 2N –darkcyan | CBS 2499 | 521 | 58 | 0.972 [0.789–1.455] | 0.239 [0.221–0.372] | 1.364 [1.301–1.804] | 0.024 [0.024–0.039] |
| Wine/Kombucha 2N –lightgreen | L14165 (UCD 2399) | 69 | 50 | 3.732 [3.13–3.732] | 0.954 [0.911–0.967] | 31.53 [16.184–31.53] | 0.631 [0.495–0.796] |
| Wine/Beer 3N –orange | AWRI1608 | 229 | 88 | 2.92 [2.557–3.65] | 0.652 [0.618–0.831] | 4.373 [3.624–14.466] | 0.05 [0.05–0.189] |
| 1st Wine 3N –red | AWRI1499 | 551 | 118 | 1.83 [1.6–2.22] | 0.384 [0.368–0.493] | 1.884 [1.776–2.581] | 0.016 [0.016–0.029] |
| Tequila/Bioethanol 3N –darkblue | CBS 5512 | 1 | 1 | NR | NR | NR | NR |
| 2nd Wine 3N –turquoise | L0308 | 40 | 26 | 2.856 [2.098–2.856] | 0.877 [0.793–0.943] | 9.756 [5.08–13.92] | 0.375 [0.323–0.708] |

For each diversity parameter, the 95% confidence interval is indicated in brackets. NR means Not Relevant.

Within these groups, contrasting genetic diversity was highlighted, with Shannon's diversity index ranging from low (0.97) to high (3.73, see Table 1). The lowest diversity was estimated for the Wine 2N group, suggesting high clonal expansion within this group, whereas higher diversity was obtained for the Wine/Kombucha 2N group and 2nd Wine 3N and Wine/beer 3N groups. Wine/Kombucha 2N and 2nd Wine 3N groups showed an equitability index closed to 1, suggesting a more even distribution of the genotypes among the genetic groups compared to Wine 2N group. Simpson's diversity and Equitability indexes showed the same trend. Overall, the percentage of the putative triploid wine isolates was 58%, indicating that the triploid state, far from being rare, has a large extend.

The genetic distribution of the wine isolates was then assessed per country, or per wine-producing region when sufficient isolates were available (Fig 2). In France, 5 regions were examined (Fig 2A): Bordeaux, Languedoc, Burgundy, Jura and Cotes-du-Rhone. In Bordeaux, 732 isolates were genotyped and were mainly distributed into two genetic groups: the Wine 2N group (darkcyan, sensitive to $SO_2$, encompassing 345/732 of Bordeaux isolates), and the 1st Wine 3N (red, tolerant to $SO_2$, 373/732). By contrast, in Burgundy, only a small percentage of the 157 isolates belonged to the Wine 2N group (16/157), while the most represented groups were the Beer 3N (orange, 95/157) and the 1st Wine 3N (red, 42/157). Cotes-du-Rhone also displayed a high proportion of isolates belonging to the Beer 3N group (orange, 26/36), beside to the Wine 2N (darkcyan, 6/36) and the 1st Wine 3N (red, 4/36). In Jura, two genetic groups dominated: the 1st Wine 3N (red, 8/16) and the Beer 3N (orange, 8/16). Finally, isolates from Languedoc mostly fell within the Wine 2N group (darkcyan 63/108), the remaining isolates belonging to the 1st Wine 3N group (red, 19/108), the 2nd Wine 3N group (turquoise, 15/108) and the Kombucha 2N group (green, 9/108). In Italy, the three regions tested (Calabria, Campania, Puglia) showed various genetic distributions, Puglia being mostly associated with the Wine 2N group, and Calabria/Campania with the 1st Wine 3N group (red). Denmark was associated with Wine 2N (darkcyan) and Beer 3N groups (orange), while Portugal showed an almost perfect equitable distribution into the five genetic groups. Isolates from Spain (mostly from Catalonia) showed the dominance of the orange group while Greece was mostly associated with the Kombucha 2N group (green) and then with the 1st Wine 3N group (red, Fig 2A). In non-European countries (Fig 2B), the genetic distribution of *B. bruxellensis* was also

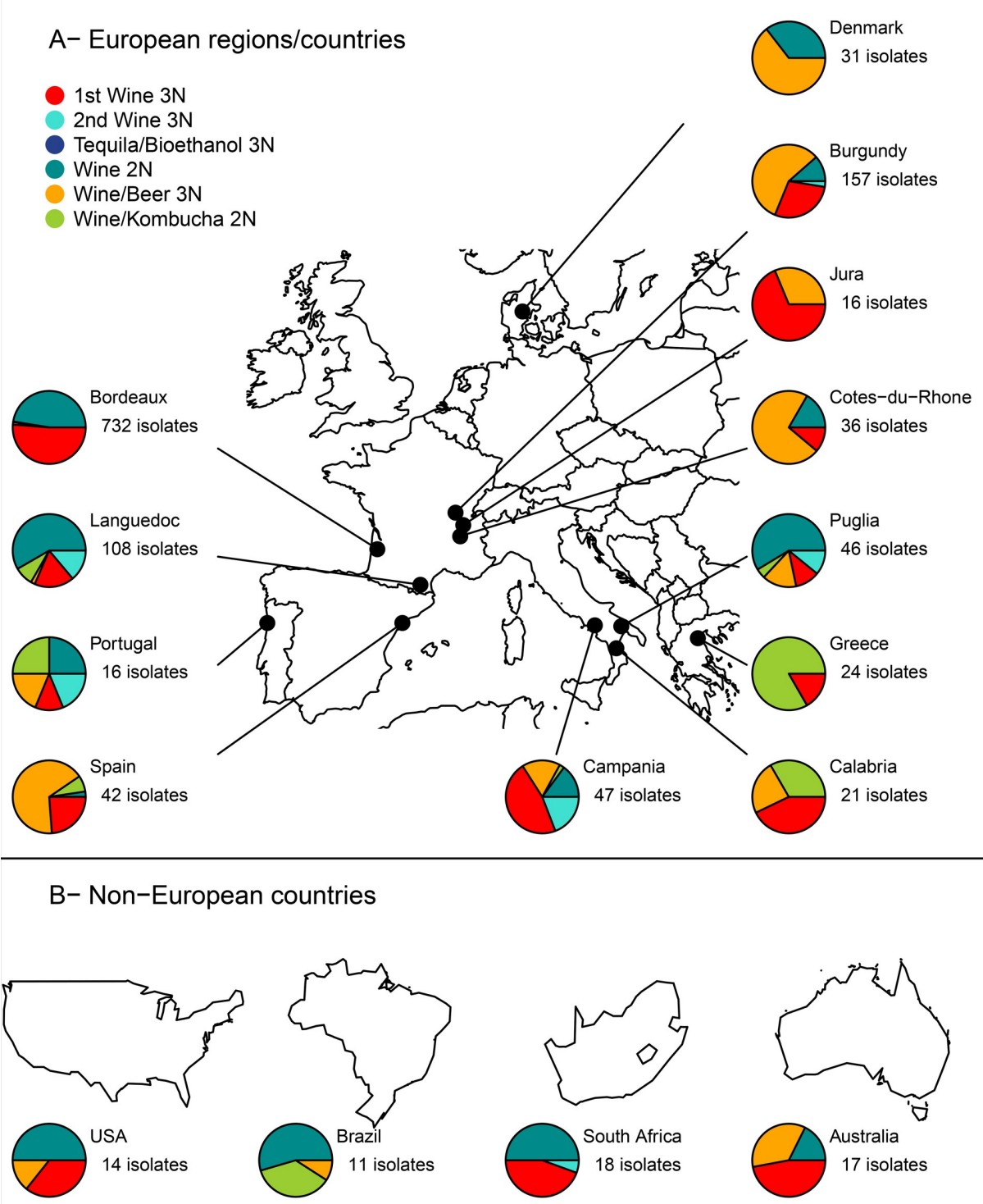

**Fig 2. Genetic distribution of *Brettanomyces bruxellensis* wine isolates in different regions or countries.** Maps were drawn using *maps* packages and pies using *graphics* package.

contrasted, with the diploid wine group (darkcyan) dominant in USA, Brazil and South Africa, and the 1st Wine 3N group (red) dominant in Australia.

When summing-up all these regional specific distributions, some trends emerged: the Wine 2N (darkcyan) and the 1st Wine 3N (red) groups were isolated in almost every region/country. The Beer 3N group (orange) was more dominant around a meridian crossing Denmark, the east of France and Italy (except for Spain), while the Kombucha 2N group (green) was mostly found around Mediterranean countries. Furthermore, isolates tolerant/resistant to sulfites (belonging to the 1st or 2nd Wine 3N groups), were found in almost all regions/countries.

## The temporal distribution of *B. bruxellensis* wine isolates reveals important evolution over the last century

Most of the studied strains were isolated in the last decade from wines sampled within two years after grapes harvesting (vintage). However, 157 isolates were isolated prior to 2000 and/or were isolated from bottles containing old vintages, mostly from Bordeaux region. For example, three strains were isolated from 1909-wine, five isolates from 1911-wine, etc. Most of the wines with vintages older than 2000 were analyzed several years after bottling (S1 Table).

The genetic distribution of *B. bruxellensis* strains in wines older than one century, with 20-years intervals, is shown on Fig 3. Without exceptions, isolates from wines produced before-1990 (104 isolates) all belonged to diploid groups, mostly from the Wine 2N group (darkcyan). The Wine 2N group, represented 67% of the isolates from wines produced in 1981–2000, and only 32% of the isolates from wines produced in 2001–2020. For the 1st Wine 3N group (red), tolerant/resistant to sulfite, the older wine displaying such isolate dates back to 1990, and was isolated 15 years after wine elaboration. The proportion of "red" isolates increased from 23% for 1981–2000 period and to 43% for the wines produced during 2001–2020. Similarly, the Beer 3N group that was first isolated from a wine of 1995 vintage represented only 4% of the isolates from wines produced between 1981 and 2000, and 18% of the isolates from wines produced between 2001–2020. For the other genetic groups, the older isolates were found in wine as old as 1956 for Kombucha 2N (and represented around 4–5% of the population), 1994 for 2nd Wine 3N (1% and 3% found in wines produced in 1981–2000 and after 2001 respectively), and 2002 for Tequila/Bioethanol (less than 1%). Unless the late sampling of the wine (>15 years) biases the analyses, the temporal distribution of *B. bruxellensis* wine isolates shows a clear shift from domination by the 2N darkcyan genetic group in old vintages to 3N red genetic group prevalence among isolates from wines produced over the last decades.

## The same *B. bruxellensis* genotypes can be identified in wines produced in a given cellar over decades

We then focused on *B. bruxellensis* clones. In this paper 'clones' will be defined as genetically identical isolates for all 12 microsatellite markers tested. Over the 1411 isolates, 138 groups of clones were identified, encompassing 2 to 114 isolates. We searched whether some clones were identified repeatedly in the same winery over different vintages. Forty-two groups of clones contained isolates isolated several times in 11 wineries from France and Italy (Fig 4). For example, in winery A1, 9 clone groups were identified: 7 from the Wine 2N and 2 from the 1st Wine 3N (red). Clones from the group n˚4 (Wine 2N) were isolated independently in wines of vintages 1909, 1948 and 1970, while clones from the group n˚8 were isolated in wines produced in 1990, 2012, 2013 and 2014. Similarly, for winery B1, 15 clone groups were evidenced: clones from the group n˚12 (Wine 2N) were isolated repeatedly in wines of vintages 1961, 1985, 1996 and 2014 wines while clones from the group n˚22 (1st Wine 3N) were isolated in

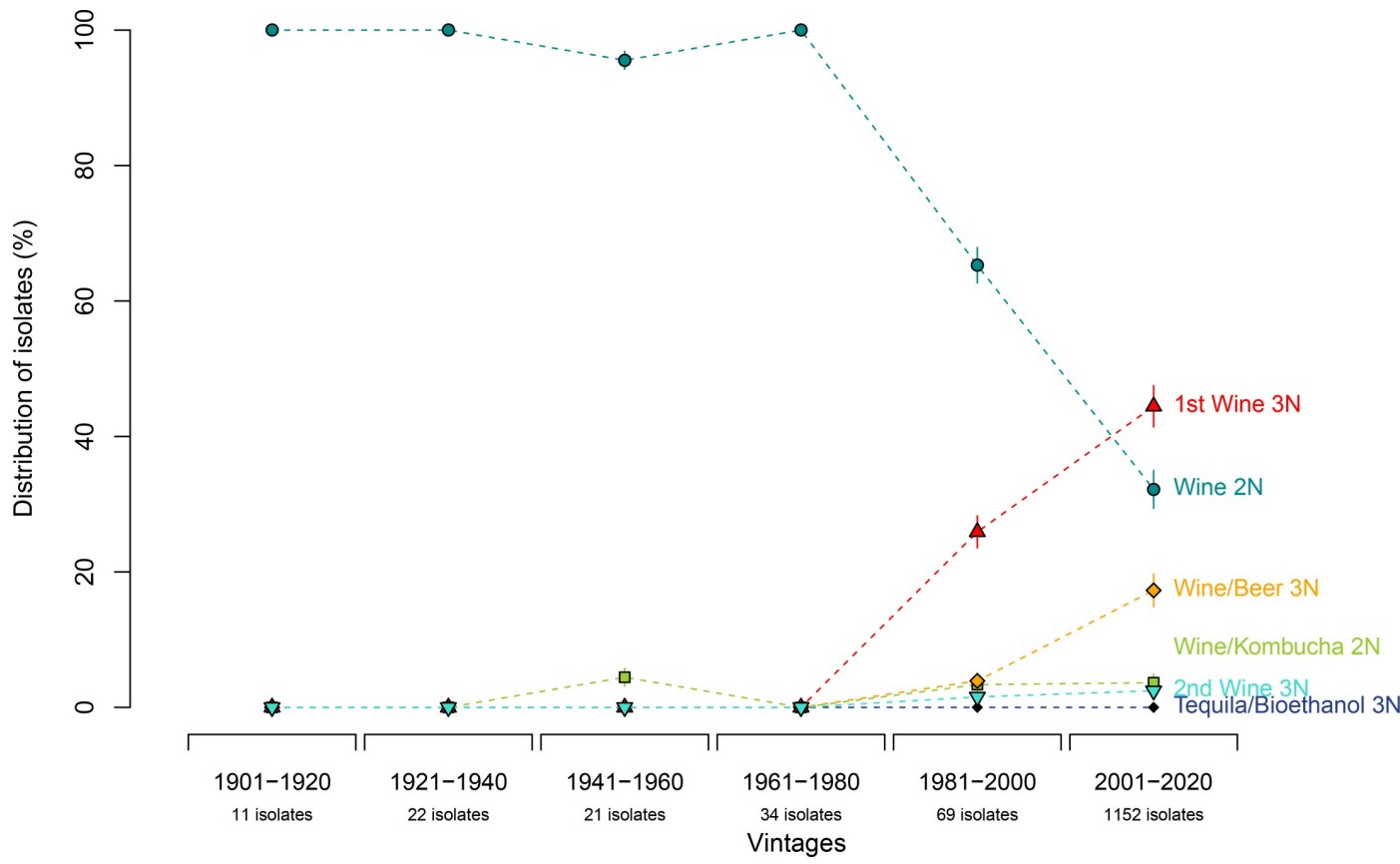

**Fig 3. Distribution of *B. bruxellensis* wine isolates from different genetic groups over vintages.** 20 years-intervals were used. In order to calculate confidence intervals, 100 bootstraps were performed (re-sampling of the population). Error bars correspond to 95% confidence interval.

wines produced in 2003, 2010, 2012, 2013 and 2014. Thus, in several wineries, genetically identical strains were isolated from wines of different vintages, sometimes from different decades. The longer interval (86 years) was found for winery B1, with clones from the group n˚3 isolated in wines produced in 1926 and 2012.

Fig 4 also illustrates that, within a given sample, different *B. bruxellensis* can be isolated. In fact, if we considered the strains isolated from the same samples, our collection contained 57 wine samples for which at least 5 isolates were analysed (mostly from France or Italy). In 45 out of 57 samples, we found isolates from two different genetic groups, highlighting the high diversity of *B. bruxellensis* at sample level.

## Wines from different countries and/or continents can be spoiled by the same *B. bruxellensis* clone

Since some clones were able to persist over several years in the cellar, we searched whether wine-producing regions were associated with specific clones. No 'signature' was identified, meaning that no specific genotypes were associated with the studied regions. Instead, we found that some clone groups were highly disseminated. For example, the clone group n˚16 (Wine 2N, darkcyan) encompassed 96 isolates from Denmark, France, Portugal and USA (Fig 5A). Another example is clone group n˚67 (6 isolates), isolated in wines from Italy, Portugal and South Africa. In the other genetic groups also, several examples of dissemination were

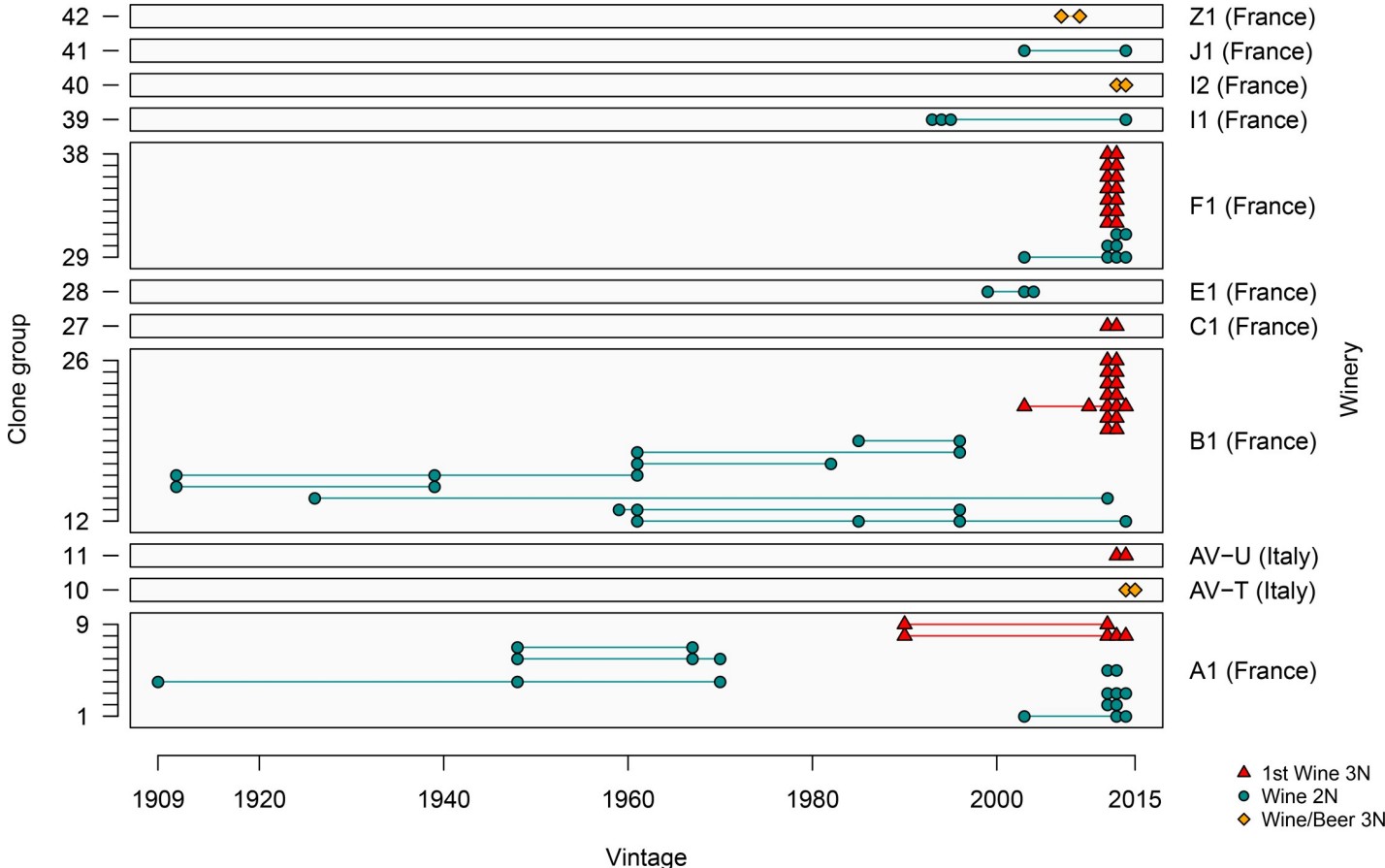

**Fig 4. Identification of clone groups within the same winery over different vintages.** Genetically identical isolates were designed as clone group. 42 groups gathering clones isolated from the same winery over different vintages were identified, corresponding to 11 wineries from France and Italy.

found (Fig 5B): the clone group n˚24 (1st Wine 3N, red) encompassed 29 isolates from France, Italy and USA, while clone group n˚35 were found in France, Italy and South Africa.

In order to quantify the level of clonal dissemination, we computed the kilometric distance separating the different isolates belonging to a given clone group (Fig 6). 88% of clone pairs were localized in the same region, with kilometric distance inferior to 100km ('local clone pairs'), of which 34% had distance inferior to 1km. 4% were separated by ~100-750km, usually associated with inter-country distances, less than 1% were distant of ~750-1000km (intra-continental distances), and 6% were separated by more than 1000km (inter-continental distances). It has to be noted that all isolates were considered here, including clonal isolates from the same wine samples that may drift the distribution toward zero kilometric distance. Thus, *B. bruxellensis* clones appear to mainly disseminate in short distances, as expected. However, a significant proportion of clones showed high geographic dispersal and these distances (hundred kilometres away) are incompatible with natural dispersion.

## Discussion

In this work, we studied the genetic diversity and structure of a large collection (>1400) of wine isolates of *B. bruxellensis*, from 21 countries across 5 continents. Most of these wine

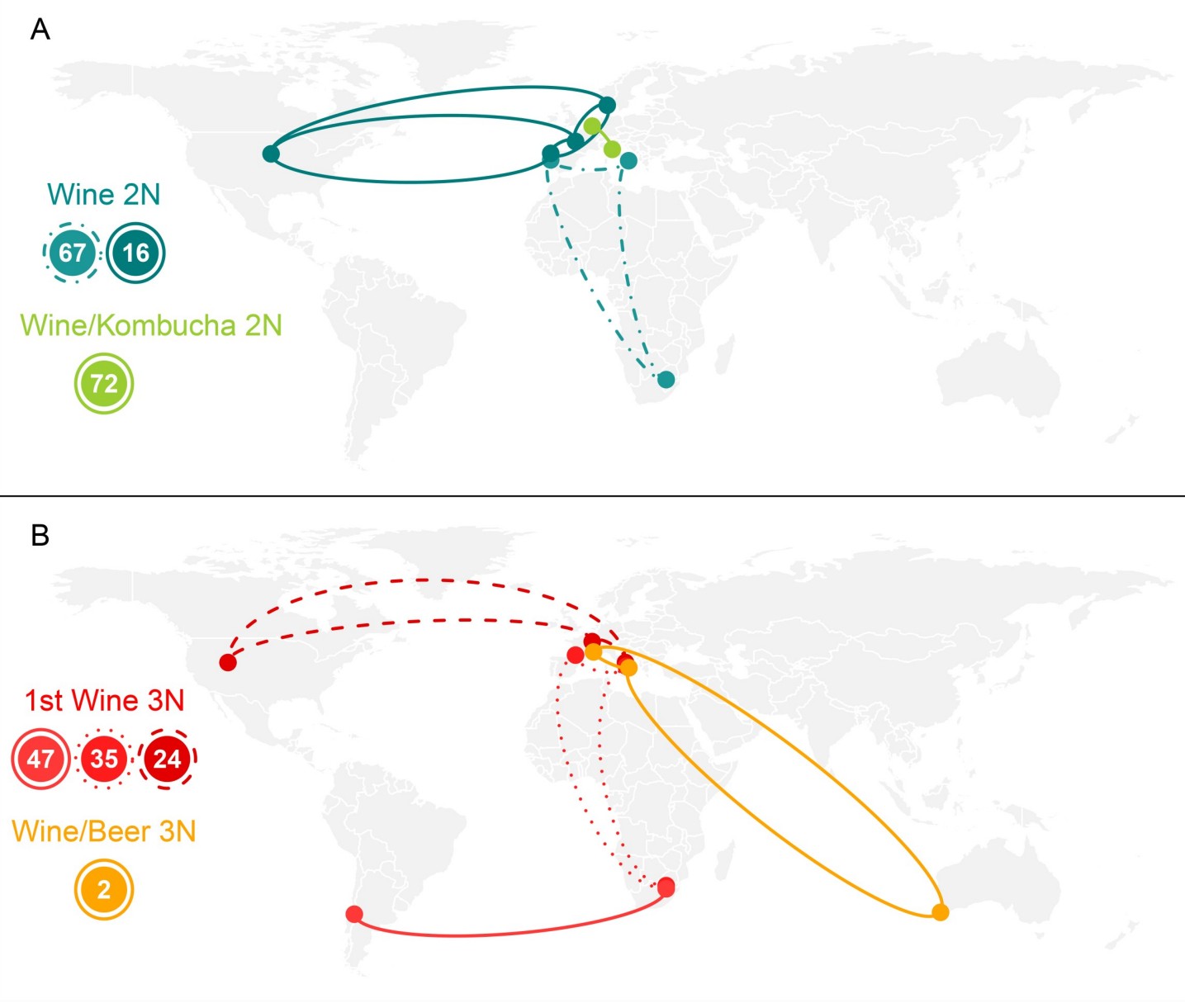

**Fig 5. Examples of spatial dispersion of wine clones of *B. bruxellensis*.** Over the 138 clone groups identified, 24 encompassed isolates from different countries. For clarity, only 7 of these groups (number 2, 16, 24, 35, 47, 67, 72) are represented here.

isolates belong to five of the six genetic groups previously described at the species level [41], and confirmed the high genetic diversity of this yeast species [23, 27]. Interestingly, we showed that the distribution of *B. bruxellensis* wine isolates varied greatly from one country/region to another. At a large scale, our results confirm that the two possible triploid groups showing sulfite resistance/tolerance are widespread worldwide and are identified in 14 regions/countries out of 16, with the notable exceptions of Denmark and Brazil. However, in both cases, the sampling may be non-representative (only 11 isolates from Brazil, and 31 isolates from a unique winery in Denmark). Indeed, for some regions, hundreds of isolates were studied (e.g. Bordeaux region, >700 isolates), while smaller subsets were considered for others (16 isolates for

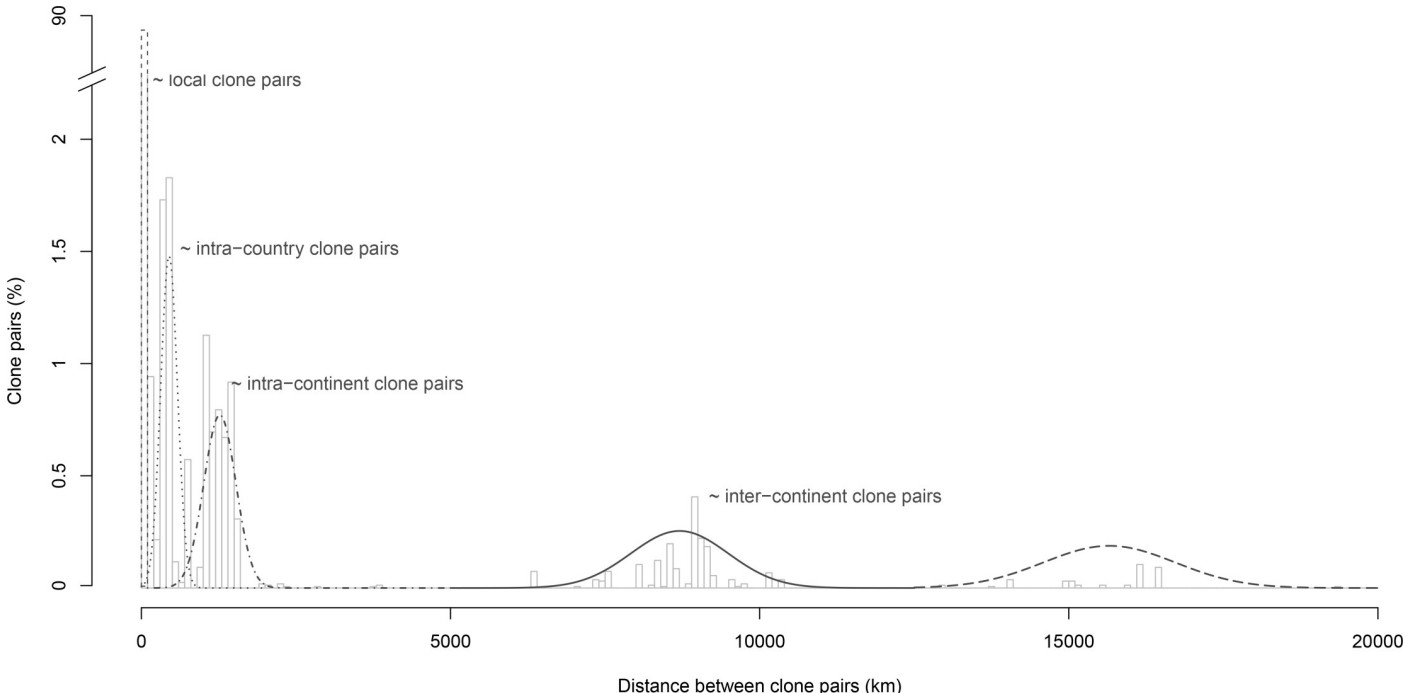

**Fig 6. Kilometric distances between wine clones of *B. bruxellensis*.** For each clone pairs, the separating distance was calculated. Genetically identical isolates separated by less than 100km were considered as "local clone pairs".

Portugal or Australia, for example). Thus, it will be necessary to validate or invalidate these results with a larger number of isolates for all regions, but also for different vintages to analyze more precisely the distribution over time for each region. At the winery level also, the number of isolates per sample (usually from 10 to 30) was low, and we can't rule out small sample size bias. Subsequent sampling of a large number of wineries, vintages and larger sample size will help refine these first results.

Still, our data reveal the unequal distribution of the different genetic groups at both the geographical and time level, suggesting that some environmental factors (climate, temperature, grape varieties etc.) leading to specific wine composition (pH, ethanol and polyphenols contents, etc.), and/or oenological practices (sulfur dioxide management, barrel ageing) could be shaping the diversity of these wine isolates. It will be interesting in a near future to identify those environmental/anthropic factors, and to examine the associated phenotypic characteristics.

## *B. bruxellensis* wine isolates show high spatiotemporal dispersion

In this paper, genetically identical isolates for all 12 microsatellites were considered as "clones". It is possible that the use of additional microsatellite markers would result in the identification of more intra-strains differences. Indeed, only full genome sequencing will assess formally whether these different isolates are actual clones. Nevertheless, the isolates hereby designed as clones, are, if not 100% identical, at least very close genetically. The analysis of these "clones" revealed unexpected patterns: first, it was observed a cellar persistence of clones over decades despite modern hygienic practices, improved cleaning/disinfection protocols and a large choice of products and treatments [52–54]. This long-term persistence of *B. bruxellensis* wine isolates in a given cellar is remarkable. In *Saccharomyces cerevisiae*, the persistence of cellar-

resident populations was shown, but on smaller period of time (over 20 years maximum) [55]. This exceptional temporal durability remains to be explored, but could be related to the specific survival ability of the species, even in a VNC form and to its bioadhesion/biofilm forming capacity in the winery environment [45, 56]. Secondly, besides its cellar residency, some *B. bruxellensis* clones showed high geographical dispersal and were independently isolated from wines originated from different producing regions, countries and sometimes even from different continents. At a regional scale, the clonal dispersal could be promoted through yeast vectors like insects and birds for a distance inferior to 100km [57–59]. However, for a significant number of clone groups, the calculated kilometric distance is incompatible with the natural dispersion, indicating the involvement of human activities. Indeed, the exchange of contaminated equipment (barrels, bottling equipment, pumps, etc.), the international wine trade and human transport of goods (fruits, etc.) could probably explain such situation [60]. The possibility to isolate clones in wines from old vintages is another example of the specific ability of the species to survive in wines after bottle aging and possibly explain world dissemination of clones through wine exports. In addition, exchanges may also happen between different industrial processes, allowing also niches dispersal of the species. However, it was previously shown that the dispersal was higher for wine isolates than other processes, suggesting different dispersal patterns for the different fermentation processes. Altogether, these results are consistent with the exchange of contaminated wine-related material, followed by adaptation to local winemaking practices, as suggested before [41]. These results draw an atypical picture of *B. bruxellensis* opportunistic lifestyle, mostly sedentary with nomad propensities.

## Allotriploidisation: a recent adaptation to winemaking practices?

One of the most interesting results of this work is the fact that isolates from old vintages mostly belong to a unique group, the so-called "wine diploid" (darkcyan), while, intriguingly, this group represents only ≈31% of nowadays isolates. The oldest isolates for the triploid genetic groups date back the 1981–2000 interval, which is particularly surprising for the 1st Wine 3N (red) group that encompasses ≈45% of recent isolates and in a less extend for the Wine/Beer 3N (orange) group showing ≈16% of recent isolates. It has to be noted that most of the 'old' isolates were actually isolated recently from old vintages (eg strain L0626 that was isolated in 2006 from a 1909 vintage). Thus, two main hypotheses can explain this result: either isolates from the Wine 2N group have higher survival or revival rates or the putative triploid groups emerged more recently, during the 1981–2000 period. It is not possible from our data to favor one or the other scenario, and the unequal sampling of the different periods (eg only 11 isolates for 1901–1920 versus 1152 isolates for 2001–2020) may bias our analysis. Subsequent strain isolations will help determine whether the different genetic groups display contrasted ability to survive in wines over decades, thus formally testing the first hypothesis. On the opposite, some elements could be consistent with the second hypothesis: first, wines produced these last decades are characterized by higher ethanol content as a consequence of climate change [61, 62]. Cibrario et al recently showed that some strains of the 1st Wine 3N (red) group were highly tolerant to high ethanol content [63]. It can be hypothesized that the progressive increase in wine alcohol level could have triggered the selection of fitter individuals regarding ethanol content. Second, the two Wine 3N groups (red and turquoise) show an outstanding phenotypic trait related to adaptation to modern winemaking practices, namely sulfite tolerance/resistance. While sulfur dioxide addition is used in winemaking at least since the 18th century, it became the preferred treatment for *B. bruxellensis* spoilage in the 90's, when Chatonnet et al. demonstrated formally that the species was the main responsible for ethylphenol production in wine [3]. Subsequently, control strategies encouraging the use of recurrent

sulfite treatments at high dosage have emerged [64]. One possible outcome of the adoption of these strategies by the wine industry might have been the selection of tolerant/resistant strains. For example, in Australia where the use of larger quantities of sulfite was promoted [60, 65], 92% of *B. bruxellensis* wine isolates were $SO_2$-tolerant in 2012 while in Greece the isolates that belong to the tolerant/resistant group were exclusively isolated from sweet red wine where higher doses of $SO_2$ are detected and permitted [66]. Winemaking environments may have supported the existence of specific selective pressure favouring the retaining of fitter allotriploid individuals and their progressive proliferation in the last decades. Indeed, competition experiments between tolerant and sensitive strains showed that the former outcompeted the latter in high $SO_2$ concentrations [44]. Altogether, our results suggest that independent allotriploidisation events in *B. bruxellensis* may have allowed diversification and subsequent adaptation to winemaking practices. Since most of the old vintages studied here were from Bordeaux region, it will be necessary to analyze old vintages from other regions to confirm or dispel such trend.

## Supporting information

**S1 Table. Details of the 1411 strains of *Brettanomyces bruxellensis* used in this study.** (CSV)

**S1 Fig. Minimum spanning tree of wine *Brettanomyces bruxellensis* isolates using a 2N-constrained dataset.** 1411 strains were genotyped using 12 microsatellite markers. For each strain (and each locus) showing 3 alleles, one of the three alleles was randomly removed to produce a randomly 2N-constrained dataset. A PCA was then performed using the R ade4 package. Only the two first axes (principal component, PC1 and PC2) were represented. The connection network and minimum spanning tree was built using the chooseCN function from R adegenet package. For genetically identical isolates (aka 'clones'), the size of the points is log10 proportional to the number of isolates. (TIF)

## Acknowledgments

The authors thank the different wineries who kindly provided wine samples. This work received financial support from the Conseil Interprofessionel des Vins de Bordeaux (CIVB, Grant number: 2014/2015 40792), from Région Aquitaine (Grant number: 2014:1R20203-00002990), from France Agrimer (Grant number: 7120154146) and by the French National Research Agency (ANR-18-CE20-0003). A.M. and M.C.P. participation was supported by a project from the Spanish Government (AGL2015-73273-JIN). The funders had no role in study design, data collection and analysis, decision to publish, or preparation of the manuscript.

## Author Contributions

**Conceptualization:** Patricia Ballestra, Warren Albertin, Isabelle Masneuf-Pomarede, Marguerite Dols-Lafargue.

**Data curation:** Alice Cibrario, Marta Avramova, Maria Dimopoulou, Maura Magani, Warren Albertin.

**Formal analysis:** Alice Cibrario, Marta Avramova, Maria Dimopoulou, Maura Magani, Cécile Miot-Sertier, Albert Mas, Maria C. Portillo, Patricia Ballestra, Warren Albertin, Isabelle Masneuf-Pomarede, Marguerite Dols-Lafargue.

**Funding acquisition:** Maria Dimopoulou, Albert Mas, Maria C. Portillo, Warren Albertin, Isabelle Masneuf-Pomarede, Marguerite Dols-Lafargue.

**Investigation:** Alice Cibrario, Marta Avramova, Maria Dimopoulou, Maura Magani, Cécile Miot-Sertier, Albert Mas, Maria C. Portillo, Patricia Ballestra, Warren Albertin, Isabelle Masneuf-Pomarede, Marguerite Dols-Lafargue.

**Methodology:** Alice Cibrario, Marta Avramova, Patricia Ballestra, Warren Albertin, Isabelle Masneuf-Pomarede, Marguerite Dols-Lafargue.

**Project administration:** Patricia Ballestra, Warren Albertin, Isabelle Masneuf-Pomarede, Marguerite Dols-Lafargue.

**Supervision:** Warren Albertin, Isabelle Masneuf-Pomarede, Marguerite Dols-Lafargue.

**Validation:** Alice Cibrario, Marta Avramova, Maria Dimopoulou, Maura Magani, Cécile Miot-Sertier, Albert Mas, Maria C. Portillo, Patricia Ballestra, Warren Albertin, Isabelle Masneuf-Pomarede, Marguerite Dols-Lafargue.

**Visualization:** Alice Cibrario, Marta Avramova, Maria Dimopoulou, Maura Magani, Cécile Miot-Sertier, Albert Mas, Maria C. Portillo, Patricia Ballestra, Warren Albertin, Isabelle Masneuf-Pomarede, Marguerite Dols-Lafargue.

**Writing – original draft:** Alice Cibrario, Marta Avramova, Maria Dimopoulou, Maura Magani, Cécile Miot-Sertier, Albert Mas, Maria C. Portillo, Patricia Ballestra, Warren Albertin, Isabelle Masneuf-Pomarede, Marguerite Dols-Lafargue.

**Writing – review & editing:** Alice Cibrario, Marta Avramova, Maria Dimopoulou, Maura Magani, Cécile Miot-Sertier, Albert Mas, Maria C. Portillo, Patricia Ballestra, Warren Albertin, Isabelle Masneuf-Pomarede, Marguerite Dols-Lafargue.

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
