## [Decision Letter · Decision Letter 0]

9 Oct 2019

PONE-D-19-25158

Brettanomyces bruxellensis wine isolates show high geographical dispersal and long remanence in cellars

PLOS ONE

Dear Dr. Albertin,

Thank you for submitting your manuscript to PLOS ONE. After careful consideration, we feel that it has merit but does not fully meet PLOS ONE’s publication criteria as it currently stands. Therefore, we invite you to submit a revised version of the manuscript that addresses the points raised during the review process.

The manuscript provides interesting and useful information about the population analysis of Brettanomyces, but some important issues are raised by both referees concerning the data, both its sources (previously published and submitted work) and its structure (different time spans between isolates). There are also issues with ploidy assessment and treatment of relevant data. Please address all points raised by both referees in your revised ms and in the rebuttal letter.

We would appreciate receiving your revised manuscript by Nov 23 2019 11:59PM. To enhance the reproducibility of your results, we recommend that if applicable you deposit your laboratory protocols in protocols.io, where a protocol can be assigned its own identifier (DOI) such that it can be cited independently in the future. For instructions see: http://journals.plos.org/plosone/s/submission-guidelines#loc-laboratory-protocols

We look forward to receiving your revised manuscript.

Kind regards,

Cecile Fairhead, Ph.D.

Academic Editor

PLOS ONE

**Journal Requirements:**

2. During your revisions, please note that a simple title correction is required: the English word 'remanence' is inappropriate in this context. Please alter the title to "Brettanomyces bruxellensis wine isolates show high geographical dispersal and long persistence in cellars", or similar. Please ensure this is updated in the manuscript file and the online submission information. Please also change the use of the word 'remanence' throughout the manuscript

**Comments to the Author**

1. Is the manuscript technically sound, and do the data support the conclusions?

Reviewer #1: Yes

Reviewer #2: Partly

2. Has the statistical analysis been performed appropriately and rigorously? 

Reviewer #1: Yes

Reviewer #2: Yes

3. Have the authors made all data underlying the findings in their manuscript fully available?

Reviewer #1: Yes

Reviewer #2: Yes

4. Is the manuscript presented in an intelligible fashion and written in standard English?

Reviewer #1: Yes

Reviewer #2: Yes

5. Review Comments to the Author

Reviewer #1: The manuscript by Cibrario and collaborators reports a population analysis of Brettanomyces bruxellensis strains carried out by means of microsatellite analysis. Thanks to this analysis, the authors found a population structure shaped by geography and strain ploidy. I addition, analysis on 100 years old wines highlighted the recent insurgence of strains resistant to sufites, suggesting a role of antropogenic activities on the evolution of the studied microbial species.

my major concern is related to the novelty of the study. Apparently, all the data were published in previous works by the same authors of this study. It seems that the vast majority of strains were characterized through microsatellite and phenotypic analyses in the work by Avramova et al. (2018). In that study, the authors reported the structure of B. bruxellensis population is shaped by geography and substrate of isolation. In the current study, they report similar resultswith an additional new result: the observation of the long-term remanence of B. bruxellensis in wines. However, the authors refer to a ‘submitted’ study which, according to the title, seems to point out exactly this result: Lleixà J, Martínez-Safont M, Magani M, Masneuf-Pomarede I, Albertin W, Mas A, et al. Genetic and phenotypic diversity of Brettanomyces bruxellensis isolates from aging wines. Food Microbiol. Submitted.

Furthermore, if different studies were collated together, how did the authors merged the microsatellite data? Microsatellite profiling tends to be highly affected by laboratory-dependent biases. Were common strains included in every analysis and used as the reference to normalize the results? In general, further details should be provided in the materials and methods section. It also strikes me the fact that diploid and triploid profiles were analyzed together. The presence of three or two alleles in some loci (as reported in supplementary table 1) is surely the main reason the strains group according to the ploidy. Unless the data were standardized before the analysis.

The most interesting finding is probably the one on ‘old wine’ isolates, suggesting an impact of antripologic activities on the selection/evolution of B. bruxellensis. However, additional factors need to be considered:

1- Have the most recent strains been isolated from must or from wines? As far as I could understand from table S1, the number of isolates from recent wines out is higher than the number of strains isolated from old wines. If this is correct, author should consider that they may have sampled only part of the population present in the wine. If all the colonies formed from the analysis of old wines were analyzed, hence all the cultivable population was sampled, author should consider that the lack of isolation of sulfite-resistant strains in old wines could also be ascribed to the fact they may be more persistent that resistant strains. Hence, future analyses of the most recent wines would result in the isolation of sulfite-sensitive strains only as the resistant one could have died earlier.

2- How do the authors explain the fact that clones were isolated from wines sampled in different years but not over the intermediate years? For instance, the third clone from winery B1 was isolated in ~1930 and in ~2014, but not in wines sampled between these two dates. This could be ascribed to either a non-representative sampling of the population or by independent infections of the same clone from external sources. In the first case, the hypothesis that strains could survive in VNC form would still stand, but the observation of the insurgence of sulfite-resistant genotypes would be based on shaking foundations as the resistant strains may not have been sampled (see also my previous comment). In the latter case, both hypotheses would still be valid, but in a different environment (not the winery).

Concerning the study on ‘clones’, authors should discuss the possibility that extending the microsatellite analysis to more than 12 loci could result in the identification of more intra-strains differences.

It is interesting that In Denmark and in Brazil sulfite-resistant strains were not isolated. It would be interesting to assess whether these two regions have something in common that makes them different from the other regions.

Minor comments:

- please use ‘structure’ rather than ‘structuration’

L47: it should be ‘published each year over the last decade’

L53: methods/approaches/analyses should be used instead of ‘markers’

L62-63: it is not clear what the authors meant here. Why genetic oddity should prompt the development of microsatellite markers? Actually, the ‘genetic oddity’ reported by the authors (existence of diploid and allotriploid strains), may rather complicate microsatellite analysis.

L76: Please check the sentence. Assessing the diversity at the species level means to compare different species, whereas the study is based on a comparison among strains.

L111: it should be the lowest diversity.

L113: please correct, it should be ‘Wine/Kombucha 2N and Wine 3N turquoise groups’

L115: please correct with ‘the genetic groups compared to...’

L236-238: please rephrase the sentence, it is not clear.

L253: a verb is probably missing here, as the verb further does not fit with the rest of the sentence.

Figure 6: are the colors used as in previous figures? If yes, why only Wine 2N are indicated as ‘inter-continenr clone pairs’? Accroding to figure 5, also clone couples of the groups red Wine 3N and Wine/Beer 3N were found in different continents. If not, please use different colors (are colors really necessary, though?)

Reviewer #2: The study by Cibrario et al. examines the geographical and temporal trends underlying Brettanomyces bruxellensis diversity in winemaking environments using previously published microsatellite genotyping data of 1411 wine isolates from 21 countries (published by Avramova et al. (2018), Dimopoulou et al. (2019) and Lleixa et al., submitted).

This study shows that B. bruxellensis wine isolates are characterized by a high genetic diversity and the distribution of the different genetic groups varies at both the geographical and temporal level. The results also show a long-term remanence of B. bruxellensis wine isolates in cellars and a high geographical dispersal among countries and sometimes even continents. One of the most interesting result of this work is the fact that the proportion of diploids and triploids changes with time. Triploids which show a higher resistance to sulphites emerged more recently and their increase in frequency is concomitant with the increased uses of sulphite treatments in the 90’s. The results suggest that allotriploidisation is a recent adaptation event to winemaking practices. This study gives insights on how the genetic diversity of B. bruxellensis could be shaped by anthropic activities. It may also be interesting for wine industry by providing better knowledge of genetic characteristics of the main wine spoiler yeast for a better spoilage prevention and improvement of treatment methods in the winery.

The manuscript is in general well-written and clear. There are some concerns that could be considered.

Major concerns:

My overarching request is that the authors better discuss and analyze some aspects of the interesting data they have.

I find the text, in places, to be somewhat assertive concerning the identification of the ploidy of strains. The microsatellite method alone is not sufficient to confirm with certitude the ploidy level of a yeast strain as we cannot exclude the presence of aneuploidies. In general, whole genome sequencing and flow cytometry (FACS) are complementary methods to confirm more accurately the ploidy level. Thus, It would be more accurate to state that the mentioned ploidy in the text is “putative ploidy” if it has been identified by microsatellite method alone.

-Figure1 : This figure shows that there is three main groups of B. bruxellensis Wine 3N, Wine /Beer3N and Wine 2N, Wine 3N and Wine/Kombucha 2N that cluster together. There is some strains that are classified in a group but cluster with another one. For example, some strains from Wine 2N cluster with the Wine/Beer 3N. It would be nice to see discussion about that.

-Figure 2 and 3: I am wondering whether the genetic distribution of B. bruxellensis wine isolates in different regions or countries presented in figure2 represent all strains isolated at the same period of time or it contains all isolates from all studied periods? The data presented in the subsequent figure 3 show that there is a variation of the genetic groups distribution over the time. It would be nice to see a geographic distribution by period of time. The same thing for the temporal distribution in figure 3, Is it the temporal distribution of all isolates from different regions and countries? It would be very interesting and informative to see the distribution of isolates by region and over the time to see the variation of a combined spatio-temporal distribution. This will shed light on the difference in evolution of isolates distribution by region/country.

-The number of isolates is very different among the different periods in figure2. The last 20 years the number of isolates is hundred times higher than the beginning of the 20th century. This should be mentioned as a limitation as made for the different number of isolates from the different regions in the first paragraph of discussion.

Minor concerns:

-Table 1 and figure 1 : There is two distinct groups called “Wine 3N”,  for clarity it would be useful to better distinguish between these two distinct groups by name and not only colours and shapes.

-Figure 2 : for clarity it would be useful to add color legend in the figure and indicate more precisely in the map the region origin of Non-European countries because countries like USA and Australia (continent) encompass a very large geographic areas.

-Figure 3 : The legend is not complete: wine/Kombucha “2N” and Tequila/Ethanol “3N”.

-Figure 4: for clarity it would be useful to add colours and shapes legends of the different genetic groups and also to indicate the geographical origins of each winery in the figure legend.

-Materials & methods : Very brief section and for the first part “Yeast strains and microsatellite genotyping”, I’m wondering if there are new genotyping in this study or all data have been already genotyped and published previously by Avramova et al., 2018 and Dimopoulou 2019 and Lleixa et al., (submitted). It is not clear why there is the Agar-YPD medium composition in this section.

-Supplementary table: the genetic group is indicated by the colour and not the group name (Wine 2N, Wine 3N… ect), for clarity and consistency, It would be better to add the group names as in the main text.

6. PLOS authors have the option to publish the peer review history of their article (what does this mean?). If published, this will include your full peer review and any attached files.

Reviewer #1: Yes: Irene Stefanini

Reviewer #2: No

---

## [Author Response · Author response to Decision Letter 0]

18 Oct 2019

Dear Editor,

Please find a revised version of our manuscript entitled ‘Brettanomyces bruxellensis wine isolates show high geographical dispersal and long persistence in cellars ’. We have carefully corrected our paper according to the reviewer’s comments, as detailed below. We would like to thank the reviewers for their constructive and insightful comments that helped to improve our manuscript.

Awaiting for your editorial decision,

Sincerely,

The authors

2. During your revisions, please note that a simple title correction is required: the English word 'remanence' is inappropriate in this context. Please alter the title to "Brettanomyces bruxellensis wine isolates show high geographical dispersal and long persistence in cellars", or similar. Please ensure this is updated in the manuscript file and the online submission information. Please also change the use of the word 'remanence' throughout the manuscript

=> Remanence was replaced by persistence throughout the text.

5. Review Comments to the Author

Reviewer #1: The manuscript by Cibrario and collaborators reports a population analysis of Brettanomyces bruxellensis strains carried out by means of microsatellite analysis. Thanks to this analysis, the authors found a population structure shaped by geography and strain ploidy. I addition, analysis on 100 years old wines highlighted the recent insurgence of strains resistant to sufites, suggesting a role of antropogenic activities on the evolution of the studied microbial species.

my major concern is related to the novelty of the study. Apparently, all the data were published in previous works by the same authors of this study. It seems that the vast majority of strains were characterized through microsatellite and phenotypic analyses in the work by Avramova et al. (2018). In that study, the authors reported the structure of B. bruxellensis population is shaped by geography and substrate of isolation. In the current study, they report similar results with an additional new result: the observation of the long-term remanence of B. bruxellensis in wines. However, the authors refer to a ‘submitted’ study which, according to the title, seems to point out exactly this result: Lleixà J, Martínez-Safont M, Magani M, Masneuf-Pomarede I, Albertin W, Mas A, et al. Genetic and phenotypic diversity of Brettanomyces bruxellensis isolates from aging wines. Food Microbiol. Submitted.

=> The study written by Lleixà J, Martínez-Safont M, Magani M, Masneuf-Pomarede I, Albertin W, Mas A, et al., entitled “Genetic and phenotypic diversity of Brettanomyces bruxellensis isolates from aging wines” that was submitted to Food Microbiol reports the phenotypic characterization of 64 Spanish isolates, that were not included in the Avramova paper. These isolates were harvested from aging wines (8-14-months aging in barrels), but not ‘old’ wines (> 2000 vintage). In Lleixa paper, microsatellite analysis is simply used to check genetic clustering before phenotypic analysis, so there is no redundancy regarding the conclusion drawn between this paper and Lleixa’s. There is no redundancy with Avramova’s paper as well, since the main conclusions of this paper (wine isolates show high geographical dispersal and long persistence in cellars) were not described in Avramova’s paper. In short, we used genotyping data from previous papers, but we described new insights through a focus on wine isolates. Note that we deleted the reference to Lleixa’s paper, since it is still in review.

Furthermore, if different studies were collated together, how did the authors merged the microsatellite data? Microsatellite profiling tends to be highly affected by laboratory-dependent biases. Were common strains included in every analysis and used as the reference to normalize the results? In general, further details should be provided in the materials and methods section. 

=> The reviewer is right; merging microsatellite data can indeed be a problem. All genotyping analyses were performed by only one lab (UR Oeno), using the same experimental conditions. But even within a given laboratory, over time, microsatellite profiling can change, although marginally (ie it is frequent to observe 1pb difference between different genotyping batches). This is why we included in our genotyping batch 1-2 reference strains for normalization. These elements were missing in the material and method section, now improved and clarified (lines 92-102 in the 'Revised Manuscript with Track Changes’).

It also strikes me the fact that diploid and triploid profiles were analyzed together. The presence of three or two alleles in some loci (as reported in supplementary table 1) is surely the main reason the strains group according to the ploidy. Unless the data were standardized before the analysis.

=> One difficulty when dealing with populations showing different ploidy level is to use appropriate analyses that will not bias the results. In a previous paper (Avramova et al, 2018), we used raw genotyping data (unstandardized regarding ploidy) and various approaches (Bruvo’s genetic distance (specifically designed to support mixed ploidy populations) and NJ clustering; UPGMA clustering, multidimensional scaling, PCA, etc. In all these cases, the clusters were globally conserved whatever the approach. In addition, a “core genotype analysis” was performed, in which alleles identified as triploid-specific were excluded in order to study specifically the “core diploid” genotypes. All analyses gave similar results and clustering, showing, using different tools/approaches, that the presence of 2 or 3 alleles wasn’t the main reason for the observed clustering. To demonstrate that point formally here, we performed the following simulation: for each strain and each locus showing 3 alleles, we randomly removed one of the three alleles and perform the same PCA analysis. The result is shown in a new figure (S1 Fig), which is very close to the PCA obtained with the complete dataset. The “Mat & Met” and “Results” sections are modified to describe the new S1 Fig (lines 109-113; 136-141 in the 'Revised Manuscript with Track Changes’).

The most interesting finding is probably the one on ‘old wine’ isolates, suggesting an impact of antripologic activities on the selection/evolution of B. bruxellensis. However, additional factors need to be considered:

1- Have the most recent strains been isolated from must or from wines? 

=> Most isolates (old or recent) were isolated from wine, scarcely any from must, lees, grape or winery environment. We added a new “Detail” column for S1 Table with these data. 

As far as I could understand from table S1, the number of isolates from recent wines out is higher than the number of strains isolated from old wines. If this is correct, author should consider that they may have sampled only part of the population present in the wine. If all the colonies formed from the analysis of old wines were analyzed, hence all the cultivable population was sampled, author should consider that the lack of isolation of sulfite-resistant strains in old wines could also be ascribed to the fact they may be more persistent that resistant strains. Hence, future analyses of the most recent wines would result in the isolation of sulfite-sensitive strains only as the resistant one could have died earlier.

=> Indeed, we can’t rule out the possibility that 2N strains are more persistent than 3N strains. This point is discussed lines 341-343 in the 'Revised Manuscript with Track Changes’. 

2- How do the authors explain the fact that clones were isolated from wines sampled in different years but not over the intermediate years? For instance, the third clone from winery B1 was isolated in ~1930 and in ~2014, but not in wines sampled between these two dates. This could be ascribed to either a non-representative sampling of the population or by independent infections of the same clone from external sources. In the first case, the hypothesis that strains could survive in VNC form would still stand, but the observation of the insurgence of sulfite-resistant genotypes would be based on shaking foundations as the resistant strains may not have been sampled (see also my previous comment). In the latter case, both hypotheses would still be valid, but in a different environment (not the winery).

=> For winery B1, 15 vintages between 1911 and 2014 were sampled, and a large diversity was evidenced (15 clone groups). Since less than 20 isolates were studied for most old samples, it may explain why the 3rd clone of B1 was found only in ~1930 and in ~2014. Thus, even if our total number of Brett isolates (>1400) is important, per sample we may have small sample size bias. This point is now discussed (lines 292-296 in the 'Revised Manuscript with Track Changes’). 

Concerning the study on ‘clones’, authors should discuss the possibility that extending the microsatellite analysis to more than 12 loci could result in the identification of more intra-strains differences.

=> Reviewer 1 is right again. This fact is now discussed (lines 304-308 in the 'Revised Manuscript with Track Changes’).

It is interesting that In Denmark and in Brazil sulfite-resistant strains were not isolated. It would be interesting to assess whether these two regions have something in common that makes them different from the other regions.

=> A small number of isolates were studied for Denmark and Brazil, prompting the need for larger samples. This point is now discussed (lines 288-289 in the 'Revised Manuscript with Track Changes’).

Minor comments:

- please use ‘structure’ rather than ‘structuration’

L47: it should be ‘published each year over the last decade’

L53: methods/approaches/analyses should be used instead of ‘markers’

L62-63: it is not clear what the authors meant here. Why genetic oddity should prompt the development of microsatellite markers? Actually, the ‘genetic oddity’ reported by the authors (existence of diploid and allotriploid strains), may rather complicate microsatellite analysis.

L76: Please check the sentence. Assessing the diversity at the species level means to compare different species, whereas the study is based on a comparison among strains.

L111: it should be the lowest diversity.

L113: please correct, it should be ‘Wine/Kombucha 2N and Wine 3N turquoise groups’

L115: please correct with ‘the genetic groups compared to...’

L236-238: please rephrase the sentence, it is not clear.

L253: a verb is probably missing here, as the verb further does not fit with the rest of the sentence.

=> We corrected the manuscript accordingly

Figure 6: are the colors used as in previous figures? If yes, why only Wine 2N are indicated as ‘inter-continenr clone pairs’? Accroding to figure 5, also clone couples of the groups red Wine 3N and Wine/Beer 3N were found in different continents. If not, please use different colors (are colors really necessary, though?)

=> Thanks for pointing this. Colors in figure 6 were indeed not related to the genetic groups, and a new figure 6 without colors is proposed.

Reviewer #2: The study by Cibrario et al. examines the geographical and temporal trends underlying Brettanomyces bruxellensis diversity in winemaking environments using previously published microsatellite genotyping data of 1411 wine isolates from 21 countries (published by Avramova et al. (2018), Dimopoulou et al. (2019) and Lleixa et al., submitted).

This study shows that B. bruxellensis wine isolates are characterized by a high genetic diversity and the distribution of the different genetic groups varies at both the geographical and temporal level. The results also show a long-term remanence of B. bruxellensis wine isolates in cellars and a high geographical dispersal among countries and sometimes even continents. One of the most interesting result of this work is the fact that the proportion of diploids and triploids changes with time. Triploids which show a higher resistance to sulphites emerged more recently and their increase in frequency is concomitant with the increased uses of sulphite treatments in the 90’s. The results suggest that allotriploidisation is a recent adaptation event to winemaking practices. This study gives insights on how the genetic diversity of B. bruxellensis could be shaped by anthropic activities. It may also be interesting for wine industry by providing better knowledge of genetic characteristics of the main wine spoiler yeast for a better spoilage prevention and improvement of treatment methods in the winery.

The manuscript is in general well-written and clear. There are some concerns that could be considered.

Major concerns:

My overarching request is that the authors better discuss and analyze some aspects of the interesting data they have.

I find the text, in places, to be somewhat assertive concerning the identification of the ploidy of strains. The microsatellite method alone is not sufficient to confirm with certitude the ploidy level of a yeast strain as we cannot exclude the presence of aneuploidies. In general, whole genome sequencing and flow cytometry (FACS) are complementary methods to confirm more accurately the ploidy level. Thus, It would be more accurate to state that the mentioned ploidy in the text is “putative ploidy” if it has been identified by microsatellite method alone.

=> We modified our text to be more prudent regarding the possible ploidy level of Brett strains (lines 28, 36, 71 and afterward in the 'Revised Manuscript with Track Changes’). However, note that, to date, our assessment of ploidy level is congruent with the one obtained using full genome sequencing. 

-Figure1 : This figure shows that there is three main groups of B. bruxellensis Wine 3N, Wine /Beer3N and Wine 2N, Wine 3N and Wine/Kombucha 2N that cluster together. There is some strains that are classified in a group but cluster with another one. For example, some strains from Wine 2N cluster with the Wine/Beer 3N. It would be nice to see discussion about that.

=> Again a pertinent comment. We kept the initial distribution into clusters performed by Avramova et al. (2018). Non-wine strains from the initial subset were removed, while some others were added, mostly wine strains from Greece (Dimopoulou et al, 2019) and Spain (Lleixa et al, submitted). The clusters were globally well conserved, except for a few strains (<15) whose position varied a bit, usually “peripheric” strains. This issue is now pointed out (lines 128-130 in the 'Revised Manuscript with Track Changes’).

-Figure 2 and 3: I am wondering whether the genetic distribution of B. bruxellensis wine isolates in different regions or countries presented in figure2 represent all strains isolated at the same period of time or it contains all isolates from all studied periods? The data presented in the subsequent figure 3 show that there is a variation of the genetic groups distribution over the time. It would be nice to see a geographic distribution by period of time. The same thing for the temporal distribution in figure 3, Is it the temporal distribution of all isolates from different regions and countries? It would be very interesting and informative to see the distribution of isolates by region and over the time to see the variation of a combined spatio-temporal distribution. This will shed light on the difference in evolution of isolates distribution by region/country.

=> Fig 2 and 3 indeed presents all wine strains (for which geographic origins and/or vintages were known). While our dataset contains an important number of wine strains, the distribution is not homogeneous over time and region, so that spatio-temporal variation can’t be assessed with this dataset. The need for more isolates from different regions and/or vintages is now discussed (lines 292-296 in the 'Revised Manuscript with Track Changes’).

-The number of isolates is very different among the different periods in figure2. The last 20 years the number of isolates is hundred times higher than the beginning of the 20th century. This should be mentioned as a limitation as made for the different number of isolates from the different regions in the first paragraph of discussion.

=> The unequal sampling of isolates over time (and over region) is now discussed (lines 344-345 in the 'Revised Manuscript with Track Changes’)

Minor concerns:

-Table 1 and figure 1 : There is two distinct groups called “Wine 3N”, for clarity it would be useful to better distinguish between these two distinct groups by name and not only colours and shapes.

=> The two Wine 3N were called 1st Wine 3N and 2nd Wine 3N respectively, in the manuscript, tables and figures.

-Figure 2 : for clarity it would be useful to add color legend in the figure and indicate more precisely in the map the region origin of Non-European countries because countries like USA and Australia (continent) encompass a very large geographic areas.

=> Unfortunately, for many strains from collection, we have no mention of the precise area of isolation (only the country). We added the legend in the Figure 2 as requested.

-Figure 3 : The legend is not complete: wine/Kombucha “2N” and Tequila/Ethanol “3N”.

=> This display issue was corrected.

-Figure 4: for clarity it would be useful to add colours and shapes legends of the different genetic groups and also to indicate the geographical origins of each winery in the figure legend.

=> The legend was added

-Materials & methods : Very brief section and for the first part “Yeast strains and microsatellite genotyping”, I’m wondering if there are new genotyping in this study or all data have been already genotyped and published previously by Avramova et al., 2018 and Dimopoulou 2019 and Lleixa et al., (submitted). It is not clear why there is the Agar-YPD medium composition in this section. 

=> This part was rewritten for clarity

-Supplementary table: the genetic group is indicated by the colour and not the group name (Wine 2N, Wine 3N… ect), for clarity and consistency, It would be better to add the group names as in the main text. 

=> The table was corrected accordingly

---

## [Decision Letter · Decision Letter 1]

14 Nov 2019

Brettanomyces bruxellensis wine isolates show high geographical dispersal and long persistence in cellars

PONE-D-19-25158R1

Dear Dr. Albertin,

We are pleased to inform you that your manuscript has been judged scientifically suitable for publication and will be formally accepted for publication once it complies with all outstanding technical requirements.

With kind regards,

Cecile Fairhead, Ph.D.

Academic Editor

PLOS ONE

Additional Editor Comments (optional):

Reviewers' comments:

Reviewer's Responses to Questions

**Comments to the Author**

1. If the authors have adequately addressed your comments raised in a previous round of review and you feel that this manuscript is now acceptable for publication, you may indicate that here to bypass the “Comments to the Author” section, enter your conflict of interest statement in the “Confidential to Editor” section, and submit your "Accept" recommendation.

Reviewer #1: All comments have been addressed

Reviewer #2: All comments have been addressed

2. Is the manuscript technically sound, and do the data support the conclusions?

Reviewer #1: Yes

Reviewer #2: Yes

3. Has the statistical analysis been performed appropriately and rigorously? 

Reviewer #1: Yes

Reviewer #2: Yes

4. Have the authors made all data underlying the findings in their manuscript fully available?

Reviewer #1: Yes

Reviewer #2: Yes

5. Is the manuscript presented in an intelligible fashion and written in standard English?

Reviewer #1: Yes

Reviewer #2: Yes

6. Review Comments to the Author

Reviewer #1: please be careful with the special character, all the Greek characters (e.g. mu for micro) have disappeared after the conversion to pdf.

Reviewer #2: The authors have thoroughly revised the manuscript and addressed each of my comments in their response letter.

I am satisfied with the majority of revisions they have made in response to my initial comments. I have only a follow-up comment that pertain to the revision. In figure 3, if I understood well and according to the supplementary table all old isolates sampled before 2000 are from Bordeau. Also, 586 of isolates sampled after 2000 are from Bordeau Thus for consistency, It would be more accurate to keep only samples from Bordeau for the temporal distribution at all time points in the figure.

7. PLOS authors have the option to publish the peer review history of their article (what does this mean?). If published, this will include your full peer review and any attached files.

Reviewer #1: Yes: Irene Stefanini

Reviewer #2: No

---

## [Editor Report · Acceptance letter]

19 Nov 2019

PONE-D-19-25158R1 

Brettanomyces bruxellensis wine isolates show high geographical dispersal and long persistence in cellars 

Dear Dr. Albertin:

I am pleased to inform you that your manuscript has been deemed suitable for publication in PLOS ONE. Congratulations! Your manuscript is now with our production department. 

With kind regards,

on behalf of

Pr Cecile Fairhead 

Academic Editor

PLOS ONE